# Systematic Literature Review and Early Benefit of Cochlear Implantation in Two Pediatric Auditory Neuropathy Cases

**DOI:** 10.3390/jpm13050848

**Published:** 2023-05-17

**Authors:** Thomas Keintzel, Tobias Raffelsberger, Lisa Niederwanger, Gina Gundacker, Thomas Rasse

**Affiliations:** 1Department of Otorhinolaryngology, Klinikum Wels-Grieskirchen, 4600 Wels, Austria; tobias.raffelsberger@klinikum-wegr.at (T.R.); lisa.niederwanger@klinikum-wegr.at (L.N.);; 2Department Health and Rehabilitation Engineering, University of Applied Sciences Technikum Wien, 1200 Vienna, Austria; gina_gundacker@outlook.com

**Keywords:** auditory neuropathy spectrum disorder, speech understanding, speech development, daily wearing time

## Abstract

Approximately 1 in 10 children with hearing loss is affected by auditory neuropathy spectrum disorder (ANSD). People who have ANSD usually have great difficulty understanding speech or communicating. However, it is possible for these patients to have audiograms that may indicate profound hearing loss up to normal hearing. This disorder is prognosed with positive, intact or present otoacoustic emissions (OAE) and/or cochlear microphonics (CM) as well as abnormal or absent auditory brainstem responses (ABR). Treatment methods include conventional hearing aids as well as cochlear implants. Cochlear implants (CI) usually promise better speech understanding for ANSD patients. We performed a systematic literature review aiming to show what improvements can effectively be achieved with cochlear implants in children with ANSD and compare this with our experience with two cases of ANSD implanted at our clinic. The retrospective review of two young CI patients diagnosed with ANSD during infancy demonstrated improvements over time in speech development communicated by their parents.

## 1. Introduction

Globally, more than 1.5 billion people experience some degree of hearing loss (HL). Of these, an estimated 466 million adults and 34 million children have hearing loss of moderate or higher severity in the better hearing ear [1]. According to a range of studies and surveys conducted in different countries, around 0.5 to 5 in every 1000 neonates and infants have congenital or early childhood-onset sensorineural deafness or severe-to-profound hearing impairment. Out of these, auditory neuropathy spectrum disorder (ANSD) accounts for approximately 8% of newly identified cases [2], with the reported prevalence values ranging from 0.5 to 19% [3], which is approximated 1 in 10 children with hearing loss being affected [4].

Auditory neuropathy spectrum disorder (ANSD) occurs when the inner ear can perceive sound, but there is trouble transmitting that information to the brain. People of all ages, from infants to adults, are receptive [5,6,7,8].

ANSD is identified via present otoacoustic emissions (OAE) and/or cochlear microphonic (CM), and absent or grossly abnormal auditory brainstem response (ABR) [9]. The main risk factors for ANSD in neonates, generally similar to those for SNHL, include positive family history, extreme prematurity, low birth weight, hyperbilirubinemia, hypoxia, birth asphyxia, exposure to ototoxic medications, and intracranial haemorrhages including intraventricular haemorrhage [5]. Additionally, ANSD has been reported in both syndromic and non-syndromic genetic forms [10,11,12] and in mitochondrial and metabolic disorders [13,14,15].

Generally, it is well accepted that untreated hearing loss leads to delayed development of speech, language and cognitive skills, which may result in slow learning and difficulty progressing in school [16]. Therefore, early detection, nowadays commonly via newborn-hearing screenings, is a vitally important element in providing appropriate support for deaf and hearing-impaired infants. Only early detection and consecutive treatment can ensure equal opportunities in society alongside normal-hearing children.

Cochlear implants (CI) usually promise better speech understanding for ANSD patients than hearing aids [3,6,7]. Furthermore, the remarkable outcomes following CI in children with sensorineural HL (SNHL) or ANSD led to further expansion of implantation criteria and advances in surgical techniques, coding strategies, and the electrode array [12,17,18,19,20]. Therefore, more children with substantial residual hearing, particularly at low frequencies, are now considered candidates for implantation.

Several reports showed that cochlear implantation in children with ANSD resulted in open-set speech perception abilities and progressive improvements in communication skills even up to being able to have telephone conversations [21,22,23,24]. This is in contrast to a few citations showing only limited benefits after cochlear implantation. We therefore performed a systematic literature review aiming to show what improvements can effectively be achieved with cochlear implants in children with ANSD.

## 2. Materials and Methods

### 2.1. Systematic Literature Review: Literature Search

Based on the identified PICOS, presented in Table 1, studies were searched. Titles and abstracts of the publications were examined to exclude unsuitable publications in the first screening. Subsequently, the full texts of the remaining papers were reviewed and excluded if the inclusion criteria were not fulfilled.

To identify all publications for cochlear implantation in ANSD, the PubMed database was consulted. The search terms and the filters applied in the process of the search are shown in Table 2. All publications up to 1 July 2022 were included.

When necessary, mean values and standard deviations were evaluated from tables or approximated from figures. If this was not possible, these publications were excluded due to lacking sufficient information for the evaluation.

The Oxford Level of Evidence Chart (https://www.cebm.ox.ac.uk/resources/levels-of-evidence/ocebm-levels-of-evidence (accessed on 1 July 2022)) was applied to rate the identified publications. Additionally, any conflicts of interest were reported considering that there could have been bias in the results. Furthermore, it was also indicated whether the studies were supported financially.

#### 2.1.1. Assessments

An excerpt of the assessments applied in the publications is described here.

##### Audiometric Assessments

**PTA (Pure-Tone-Audiometry):** It is possible to test a person’s hearing sensitivity to calibrated pure tones utilizing pure-tone threshold audiometry [20]. The hearing threshold represents the minimum sound intensity that an ear can detect as an average of typical test frequencies in the better ear [16].**SRT (Speech Recognition Threshold):** The SRT is the minimum level at which a person can recognize 50% of two-syllable words from a closed list. The person should repeat or in some other way indicate recognition of the speech material 50% of the time. The words are presented from the front to the respondent. In the cases presented in this review, the child’s realistic index of the Speech Perception Jr. (CRISP Jr.) test was utilized for the SRT evaluation [22,23].**CAEP (cortical auditory evoked potential):** P1 latency is given in milliseconds and examines the state of the central auditory pathway’s development; furthermore, it has been applied to assess how the maturation of the auditory pathway has changed in congenitally deaf children after they have been treated with hearing aids or cochlear implants [24].

##### Speech Perception

**PBK (Phonetically Balanced Kindergarten) word list:** Word recognition assessed with the PBK word list. This test consists of a list of 50 monosyllabic words. Therefore, the test items must be repeated by the subject. The responses are evaluated by the percentage of words and phonemes correctly identified [21].**NU-CHIPS (Northwestern University—Children’s Perception of Speech):** It is applicable for kids younger than three years. It is a closed-set picture-pointing word recognition exam; 50 words that should be known to three-year-olds are spread across the test [25].**HSM (Hochmair–Schulz–Moser) sentence tests:** The HSM sentence exam is made up of 30 lists, each comprising a total of 106 words in 20 common sentences of three to eight words. The test was created with the intention to assess cochlear implant users’ speech understanding. Adding noise makes the task more difficult. Users are able to achieve 0% to 100% understanding of the words by repeating them [26].This review also presents the results of other assessments for speech recognition, which are similar in principle to the tests already explained. The following scales and word or sentence tests have been performed:Scales: The speech perception category score is based on achievements in other speech perception assessments such as the MLNT (Multisyllabic Lexical Neighbourhood Test) or closed-set speech perception test [27]. The MSP (Melbourne Speech Perception) score with a scale ranging from 0 to 7 is a similar assessment [28].Word/sentence tests: CID (Central Institute of Deafness) test [21]; MWT (Monosyllabic Word Test) [27,28]; AZBio sentence test [29]; BKB (Bamford–Kowal–Bench) sentence test [30]; MAPTB (Mandarin Auditory Perception Test Battery) [31]; CVC (Consonant–Vowel–Consonant) words [32]; CNC (Consonant–Nucleus–Consonant) words [33]; and ESP (Early Speech Perception) category [34].

##### Questionnaires

**IT-MAIS (Infant–Toddler Meaningful Auditory Integration Scale):** It is a systematic interview for the parents, created to evaluate the child’s spontaneous reactions to sounds in their daily environment. The major areas are vocalization habits, sound detection and sound interpretation. Performance is evaluated based on the overall sum of points earned out of a maximum score of 40 points. Each question has a possible score ranging from 0 (lowest) to 4 (highest) [35].**MUSS (Meaningful Use of Speech Scale):** This is a parent-reported scale that assesses children’s language use in everyday situations. The MUSS scale estimates verbal skills categorized into five categories: vocal behaviour, family conversations, social situations, language comprehension, and verbal description skills. This assessment was developed to determine how cochlear implants can help improve speech skills in children with severe hearing impairments [36].**LittlEARS hearing questionnaire:** The LittleEARS questionnaire was developed to help professionals and caregivers monitor the hearing abilities of young children with hearing loss. It is based on parents’ observations of their children’s listening behaviour in everyday life. It contains 35 age-dependent questions that must be answered with yes, one point, or no, zero points. Therefore, in total, there are 35 points that can be achieved, which are given either in absolute or percentage terms. The questions reflect important milestones in hearing development up to two years of age [37].Other questionnaires such as the parental and salt (speech and language therapist) benefit score [38] as well as the SADL (Satisfaction with Amplification in Daily Life) score [39] were applied.

##### Scales

**CAP (Categories of Auditory Performance):** In order to provide a usable measurement for non-specialists, the CAP is a global outcome measure of the development of auditory skills in deaf children. It involves the use of established criteria and observation to evaluate how well the children perform in normal scenarios at home and at school. It covers a wide variety of abilities, including understanding common phrases and conversations without lip-reading and using a phone with a known speaker. The scale ranges from 0, no awareness of ambient noise, to 7, where even the use of a phone with a known listener is possible [40,41].**SIR (Speech Intelligibility Rating):** The Speech Intelligibility Rating classifies the intelligibility of the children’s speech while using cochlear implants or hearing aids. SIR is a five-point hierarchical scale that describes different levels of speech intelligibility, from level 1, speech that is incomprehensible, to level 5, speech that is understood by every listener [40,41].

### 2.2. Our Experience with ANSD and Cochlear Implantation

In Austria, a nationwide universal newborn hearing screening is in place. Most ANSD cases, such as the two cases described here, arise from this screening program, based on OAE for low-risk and ABR measures for high-risk infants. When in the subsequent diagnostic assessment, the evaluation of ABR waveforms showed inconsistent ABR, OAE and/or CM data are categorized as possible ANSD. Following identification as ANSD, children are referred to auditory verbal therapy and will have regular audiological assessments.

The two patients were referred from the newborn hearing screening program for diagnostic audiologic evaluation and the clinical finding of ANSD was initially confirmed by (1) abnormal ABRs, (2) preserved CM responses, and (3) recordable distortion product otoacoustic emissions (DPOAEs). Both patients received a low or insufficient benefit from hearing aids and were thus considered for cochlear implantation.

The patients underwent pre-operative acoustic steady-state response testing (ASSR) and pure tone measurements post-operatively every 3 months up to 27 months (+/−2 months depending on availability). The Auditory Performance and Speech Intelligibility rating was given by the parents and evaluated by the speech therapists one year after switch-on.

## 3. Results

### 3.1. Systematic Review

#### 3.1.1. Literature Search

A total of 222 publications were identified and taken into consideration for the first screening process. The title and abstract screening excluded 34 publications. In this stage, most publications were excluded because the treatment in demand was not included, or it was not about the intended population. All reasons are presented in the flow diagram of the study selection in Figure 1. For the remaining 188 publications, full-text screening was performed. In this process, 130 articles were excluded. In this round, most of the papers were omitted since no outcomes were reported. This includes summaries on the topic and editorials but also publications that only provide graphical wave presentations such as the ABR waves in the study by Runge Samuelson et al., 2008 [42]. Unfortunately, some studies could not be analysed further because they were only available in foreign languages such as Chinese or Russian. Nevertheless, at the end of the screening process, 58 publications were considered for the data extraction, as presented in Figure 1.

#### 3.1.2. Outcomes

Figure 2 presents all extracted data from the included publications. The results vary from standard audiological tests such as the PTA to speech comprehension outcomes or statements about quality of life or quality of hearing. It also indicates which of the identifying parameters of auditory neuropathy spectrum disorder were present in the study cohorts. This means whether the auditory brainstem response was abnormal or absent as well as if otoacoustic emissions and/or cochlear microphonics were present.

In most publications, it was stated that there was a trial period with a conventional hearing aid before cochlear implantation, because this was part of the indications to be considered for a CI in the first place [6,7,8,12,21,29,31,32,33,34,44,45,46,47,48,49,50,51,52,53,54,55,56,57,58,59,60,61] with ANSD. If there was no improvement in speech understanding or language development with these interventions, the children were fitted with a CI. The most common tests in the extracted studies are the pure tone average, the CAP and SIR category score, but also the IT-MAIS questionnaire, which is administered to the children by their parents. As these scores are the most frequently performed, they have been summarised in Table 3 as mean scores for the pre- and post-operative time points.

A total of 919 children who were diagnosed with auditory neuropathy spectrum disorder and subsequently received a cochlear implant were examined in the publications. Implants from the following manufacturers were applied: Advanced Bionics, Cochlear, MED-EL and Nurotron. As already mentioned, the patients discussed are children, but in the study by Gibson et al., 2007 [23], no age was given; nevertheless, it was emphasised that the study cohort consisted of children.

Most of the extracted publications were case reports or consisted of small study cohorts. However, the studies [6,45,62,63] included over 50 children. The most common risk factors for ANSD were hyperbilirubinemia, prematurity, jaundice, NICU (= neonatal intensive care unit), ototoxic medication or ear infections. In some cases [52,61,64,65,66], an OTOF gene mutation was also named as a trigger for auditory neuropathy. Although not often mentioned but present in some publications [31,57,60], ANSD patients had a normal tympanic membrane, normal auditory nerves and normal middle- and inner-ear structures. This was determined with the help of MRI or high-resolution CT scans.

In the studies [32,38], riboflavin therapy was applied because in these cases ANSD was associated with the Brown–Vialetto–Van Laere (BVVL) syndrome, which is characterised by a riboflavin transporter deficiency. Other therapies utilised in auditory neuropathy patients were intensive speech and auditory therapy, but mostly before the CI was applied [6,29,31,44,48,53,54,60].

**Table 3 jpm-13-00848-t003:** Mean values for CAP, SIR, IT-MAIS and PTA outcomes before and after CI from various publications.

Assessment	Timepoint	Mean	SD	Subjects	Publications
CAP	Pre-OP	0.95	0.57	190	[3,32,42,51,55,64,67,68,69]
	Post-OP	5.07	0.98	196	[3,32,42,50,51,55,60,64,67,68,69,70]
SIR	Pre-OP	1.39	0.42	166	[3,55,68,70]
	Post-OP	3.45	1.12	141	[3,50,51,55,65,67,68,70]
IT-MAIS	Pre-OP	16%	7.69%	77	[28,54,59,61,64,65,66,67,71,72]
	Post-OP	77%	14.85%	93	[28,54,59,61,64,65,66,67,68,72,73]
PTA	Pre-OP	86.7 dB	16.5 dB	301	[5,25,28,31,32,35,42,48,49,52,54,62,63,64,66,71,73,74]
	Fitted with HA	68.6 dB	16.1 dB	88	[35,49,56,61,64,65]
	Post-OP	33.7 dB	16.0 dB	241	[5,22,25,32,33,42,48,49,52,53,54,56,57,59,61,62,63,64,65,74,75,76,77,78]

**Figure 2 jpm-13-00848-f002:**
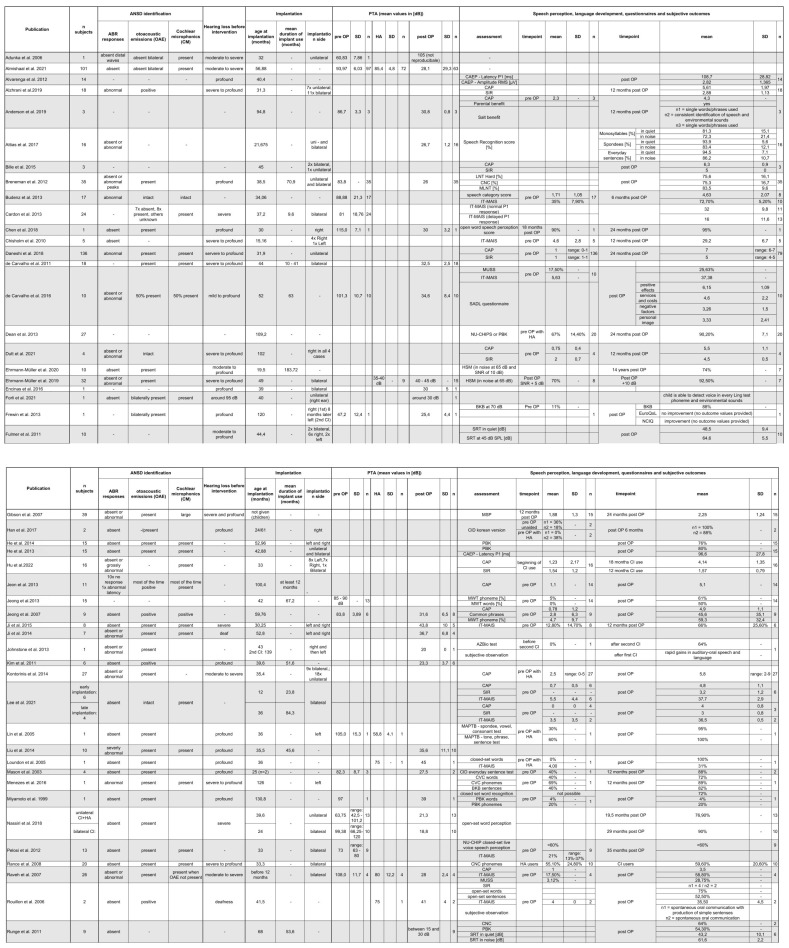
Results for all included publications. AZBio = AZBio sentence test; BKB = Bamford–Kowal–Bench sentence test; CAEP = cortical auditory evoked potential, CAP = Categories of Auditory Performance; CID = Central Institute of Deafness test; CNC = Consonant–Nucleus–Consonant words; CVC = Consonant–Vowel–Consonant words; ESP = Early Speech perception test; HA = Hearing aid; HSM = Hochmair–Schulz–Moser sentence test; IT-MAIS = Infant–Toddler Meaningful Auditory integration scale; LNT = Lexical Neighbourhood Test; MAPTB = Mandarin Auditory Perception Test Battery; MLNT = Multisyllabic Lexical Neighbourhood Test; MSP = Melbourne Speech Perception; MUSS = Meaningful Use of Speech Scale; MWT = Monosyllabic Word Test; NU-CHIPS = Northwestern University—Children’s Perception of Speech; PBK = Phonetically Balanced Kindergarten word list; SADL = Satisfaction with Amplification in Daily Life; SIR = Speech Intelligibility Rating, SRT = Speech Recognition Threshold [4,5,6,9,12,15,16,17,18,19,20,22,23,24,25,29,35,36,37,38,39,40,41,42,43,44,45,46,47,48,49,50,51,52,55,56,57,58,59,60,61,62,63,64,65,66,67,68,69,70,71,72].

Table 3 shows that different auditory assessments have been used to identify benefits or deterioration from a CI intervention in ANSD patients. As already mentioned, the most frequently listed assessments are summarised in Table 3. The IT-MAIS was given in the publications either as an absolute or percentage value. To achieve a uniform value, the absolute values in Table 3 were converted to percentages in order to be able to present a percentual mean value in Table 3.

Table 3 shows that better results were achieved in all parameters after the implantation of a cochlear implant. In the pure tone average, better hearing thresholds were also achieved with cochlear implants (33.7 dB ± 16.0 dB) than with conventional hearing aids (68.6 dB ± 16.1 dB). However, this was probably also the reason why a CI was considered as a treatment method in the first place.

The quality classification and statement on conflicts of interest or financial support are shown in Figure 3. Out of the 58 extracted citations, 31 papers reported having no conflicts of interest and 23 publications did not provide a conflict-of-interest statement. Only four papers with a conflict of interest had members who were additionally active on the review board of the hearing implant companies. In terms of financial support, 22 publications stated that they received support from the state or research funds. In 22 papers, no information was given, and 14 reported that they had not received any financial support. The extracted papers were rated with the Oxford level of evidence. Two-thirds of the papers were classified as class IV and IV to V. The low rating of the publications is presumably because many case reports were included, which was also the case with the ten class V studies. Only two retrospective studies could be classified as level III. Four publications reached the intermediate level III to IV.

### 3.2. Case Presentations

Two ANSD cases reported to our clinic, one female and one male aged 8 months and 25 months at the time of diagnosis, respectively. An MRI of the cerebrum and a CT of the petrous bones were performed for both cases, indicating no abnormal anatomical condition.

Both children also underwent genetic testing, which revealed a mutation in the OTOF gene, indicating the possibility of an autosomal recessive hearing loss type 9/auditory neuropathy type 1. These results pointed towards the diagnosis of auditory neuropathy.

Case 1 was a male boy who trialed other options for 7 months before opting for an implant. Case 2 was a girl who had an implant fitted 2 months after presenting to our clinic as no other rehabilitation option would be tolerated by the child.

#### 3.2.1. Case 1

A male infant presented to our clinic with a diagnosis of ANSD, following abnormal ABR testing with preserved CM responses, and recordable distortion product otoacoustic emissions (DPOAEs). The parents reported no benefit from hearing aid trials and thus considered cochlear implantation for their child. At the age of 15 months, the boy was implanted with a Synchrony 2 implant and a Flex Soft array (MED-EL) on both sides simultaneously. Surgery was uneventful and activation/first fitting of the Sonnet 2 Audio Processor (MED-EL) was performed 24 days after surgery.

The subject underwent pre-operative acoustic steady-state response testing (ASSR) and pure tone measurements post-operatively every 3 months up to 27 months (+/−2 months depending on availability) (Figure 4). The Auditory Performance and Speech Intelligibility rating was given by the parents and evaluated by the speech therapists (Figure 5, right panel) one year after switch-on.

Twenty-seven months after implantation, the child uses the implant for about 11.94 ± 1.34 h per day (as extracted from manufacturer data logging). Free field threshold with CI improved from 2 months after surgery with a mean of 77.5 dB to 41.67 dB at the latest visit (pure tone audiometry between 0.5 and 1–2–4 KHz). Despite still being averse to loud noises, the parents are very satisfied with the development of the child. The child is able to detect voices, tries to copy environmental sounds such as animals, and produces various vocalizations and a few onomatopoeias. Speech audiometry was conducted but did not result in useful outcomes due to a lack of concentration; nonetheless, the speech understanding as reported by the mother is very good.

#### 3.2.2. Case 2

A 25-month-old female infant presented to our clinic with a diagnosis of ANSD, following abnormal ABR testing with preserved CM responses and recordable DPOAEs. The parents reported no acceptance of hearing aids and thus considered cochlear implantation for their child within 2 months of diagnosis. At the age of 27 months, the girl was bilaterally provided with a Synchrony 2 implant and a Flex Soft array (MED-EL). Surgery was uneventful; neither intra- nor post-op complications were reported. The activation/first fitting of the Sonnet 2 Audio Processor (MED-EL) was performed 23 days after surgery.

The subject underwent pre-operative acoustic steady-state response testing (ASSR) and pure tone measurements post-operatively every 3 months up to 25 months (+/−2 months depending on availability) (Figure 6). The Auditory Performance and Speech Intelligibility rating was given by the parents and evaluated by the speech therapists (Figure 5, left panel) one year after switch-on.

**Figure 5 jpm-13-00848-f005:**
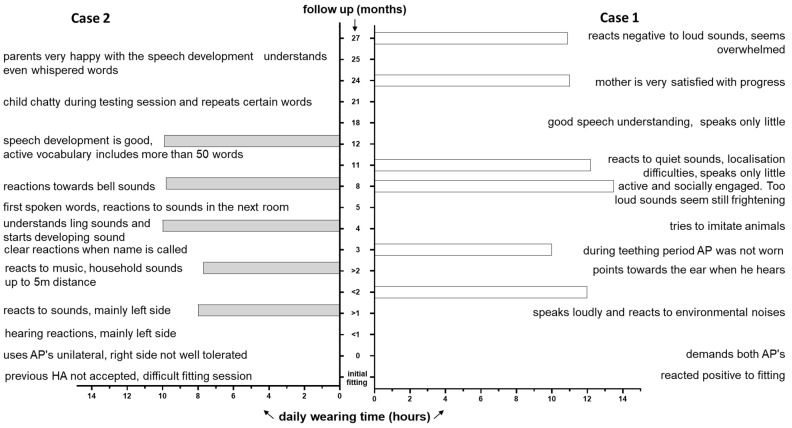
Daily wearing time (*x*-axis) combined with the parents’ and speech pathologist’s subjective impression of the child’s post-operative development. *y*-axis (centre) denotes the follow-up in months. (**right**) Case one; (**left**) Case two.

Twenty-five months after implantation, the child uses the implant for about 9.08 ± 1.01 h per day (as extracted from manufacturer datalogging).

Free field threshold with CI improved from 2 months after surgery with a mean of 70.0 dB to 38.33 and 40.0 dB (left and right side, respectively) 25 months after the intervention (pure tone audiometry between 0.5 and 1–2–4 KHz). Despite not tolerating the right-side audio processor as much as the left side, the parents are very satisfied with the development of the girl (Figure 5, left panel). The parents also report that the hearing reactions are mainly on the left side. The girl clearly responds when called and understands ling sounds. The child started with her first spoken words already 5 months after surgery. The girl demands her APs as soon as she gets up and wears the devices all waking hours (between 7.7 and 10 h per day). She is very chatty and not shy anymore, which was confirmed by the therapist at the latest fitting session. As with Case 1, speech audiometry was attempted but did not result in useful outcomes due to a lack of concentration. Certain words such as “mother” resulted in a noticeable reaction without a measurable outcome. Nonetheless, communication and conversations, especially during the home-rehabilitation training sessions, were reported by the mother as very good (Figure 7). Interestingly, the child did not react to vibrations but did to mobile phone ringtones, and she starts dancing when music is played.

The latest fitting session was performed 25 months after surgery and the child was able to answer whispered questions and understand commands, and she was very chatty, but her speech was difficult to understand. The vocabulary of the girl was 50+ words, which is roughly expected for a two-year-old. The parents are very satisfied with the steady and fast development of their child, which they did not expect after counselling regarding the relatively late implantation age (27 months) and when the diagnosis of ANSD and rehabilitation prospects were given.

## 4. Discussion

The outcomes of the performed systematic review are in line with our experiences in the two presented cases. Except for one study, all publications where the pure tone average was assessed achieved a hearing threshold of mild hearing loss to normal hearing conditions. In Table 3, for Adunka et al., 2006 [44], however, the result decreased, from 60.8 dB to 105 dB, and was not even reproducible for the patient. The reason for this could be that the measurement took place only two weeks after the cochlear implantation and it was not stated whether the implant or the audio processor had already been activated. Apart from this outlier, it is clearly visible in Table 3 that an improvement was achieved with the use of CIs for the ANSD patients. Even though it is a big decision for parents because a CI implantation is an invasive procedure, most papers [6,7,8,12,22,29,31,32,45,46,47,48,51,53,54,56,57,58,59,60,61,62,63,65,66,68,69,71,79] advocate for a CI for children with ANSD because of the improved speech recognition or sound recognition.

Some studies argued for considering hearing aids because a CI is not useful for everyone, but most of the time, these are special cases with comorbidities such as BVVL or Friedrichs ataxia. However, the study by Miyamoto et al., 1999 [21], presented an improvement in audiological word recognition with ANSD and the comorbidity Friedrichs ataxia from not even being able to conduct the measurement to a score of 72% word recognition after the CI implantation. However, speech production, which was assessed with the PBK word list, showed no enhancement with the CI. This does not necessarily mean that the CI is not useful; it may simply be that the child was not able to cope with the PBK test or that the additional illness affects the outcomes.

Table 3 also presents whether implantation was performed bilaterally or unilaterally. In general, better results were achieved for most outcomes with a CI, regardless of implantation side or whether the children were implanted bilaterally. In the study by Nassiri et al., 2018 [59], bilaterally implanted children were compared with a group of unilaterally implanted children with an additional hearing aid on the contralateral ear. In this case, the bilaterally implanted children performed better (90%; *n* = 10) than the unilaterally implanted children with an additional hearing aid (76.9%; *n* = 13) on the open-set word perception score. However, since the data collected after surgery are far apart, 20 and 29 months, this comparison is not meaningful and could yield the same outcomes over time.

The P1 latency of cortical auditory evoked potential is usually delayed in the presence of hearing loss. In the study by Alvarenga et al., 2012 [75], not only were results for the ANSD group determined but P1 latencies from other publications with normal hearing children were also gathered. For the ANSD group with 14 children examined, a P1 value of 108.7 ms was observed, just slightly exceeding the maximum of the normal-hearing children (107 ± 10 ms). The minimum is 61.0 ± 27 ms in the comparable age group. The cohort studied is therefore still within the tolerance and can show almost the same results as children with normal hearing. The children in the publication by Sarankumar et al., 2018 [79], performed even better with a P1 latency of 60.1 ± 26.4 ms. The fact that this group with the CI has similarly good values as the children with normal hearing can also be seen in the good outcomes of the CAP, SIR and IT-MAIS scores. The studies by He et al., 2013 and 2014 [76,77], with a P1 latency of 96.6 ± 27.8 ms are also still within the range of those with normal-hearing peers. In the study by Cardon et al., 2013 [68], the outcomes of the IT-MAIS were divided into two subgroups, one normal and delayed P1 latency response group, although it was unfortunately not defined exactly which times were considered for the classification. However, the ANSD children with a normal latency achieved better outcomes with the IT-MAIS (32 ± 9.8) than the children who had a delayed P1 response despite the cochlear implant (16 ± 11.6).

As already mentioned in the results, there were no uniform scores among the publications. A variety of language comprehension evaluations were utilised as well as questionnaires, making it difficult to interpret the results in a consistent manner. However, it was seen that in all assessments there were improvements for the children with ANSD who received a cochlear implant. However, it is repeatedly stated in the literature that CIs should only be applied to ANSD patients when conventional hearing aids do not provide any auditory improvement [7,8,22,67,68,69,71]. However, if CIs are used, it is recommended that they be applied as early as possible because the children then have the chance to achieve the same hearing conditions as people without ANSD [6,22,28,34,46,67,69,70,72,73,74,78,79,80].

Some comparative studies [8,22,46,71,81] not only assessed parameters for ANSD patients but also contrasted them with a control group of the same population size with sensory neural hearing loss (SNHL). For the study of Alzhrani et al., 2019 [46], the SNHL control group, who was also treated with a CI, performed similarly according to the CAP and SIR scores. At the timepoint of one year post-OP, the control group reached a CAP score of 6.1 ± 2.1 (*n* = 40) while the ANSD group provided a score of 5.6 ± 2.0 (*n* = 18). Furthermore, the SIR score after one year of the implantation was 3.4 ± 1.2 (*n* = 40) for SNHL patients and 2.9 ± 1.1 (*n* = 18) for the ANSD group. Comparative values for the pure-tone average were also provided by Attias et al., 2017 [71]. In this comparative study, the ANSD patients performed even better with a hearing threshold of 26.7 dB ± 1.2 dB (*n* = 16) than the SNHL control group with a value of 28.1 dB ± 1.4 dB (*n* = 16). The similarity of results between ANSD and SNHL groups receiving CIs is also confirmed by the study by Fulmer et al., 2011 [81]. There, the SRT was performed in quiet and in noise (at 45 dB SPL dB). The ANSD group achieved a score of 48.5 dB ± 9.4 dB (*n* = 10) in quiet and 64.6 dB ± 5.5 dB (*n* = 10) in noise while the SNHL group achieved 41.6 dB ± 6.4 dB (*n* = 10) in quiet and 61.9 dB ± 5.8 dB (*n* = 10) in noise.

In general, it can be said that in the included studies of this systematic review, children with ANSD benefited from cochlear implants, which is coherent with our experience. The use of cochlea implants in ANSD patients can lead to a clear increase in quality of life, speech comprehension and speech development.

## 5. Conclusions

We presented two young children diagnosed with ANSD that successfully received and benefited from CI. After careful and detailed evaluations, audiological training follow-ups and tailored rehabilitation plans, patients should be considered as beneficial candidates for the implantation of cochlear implants, especially with the diagnosis of auditory neuropathy. The use of cochlea implants in ANSD patients can lead to a clear increase in quality of life, speech comprehension and speech development.

## Figures and Tables

**Figure 1 jpm-13-00848-f001:**
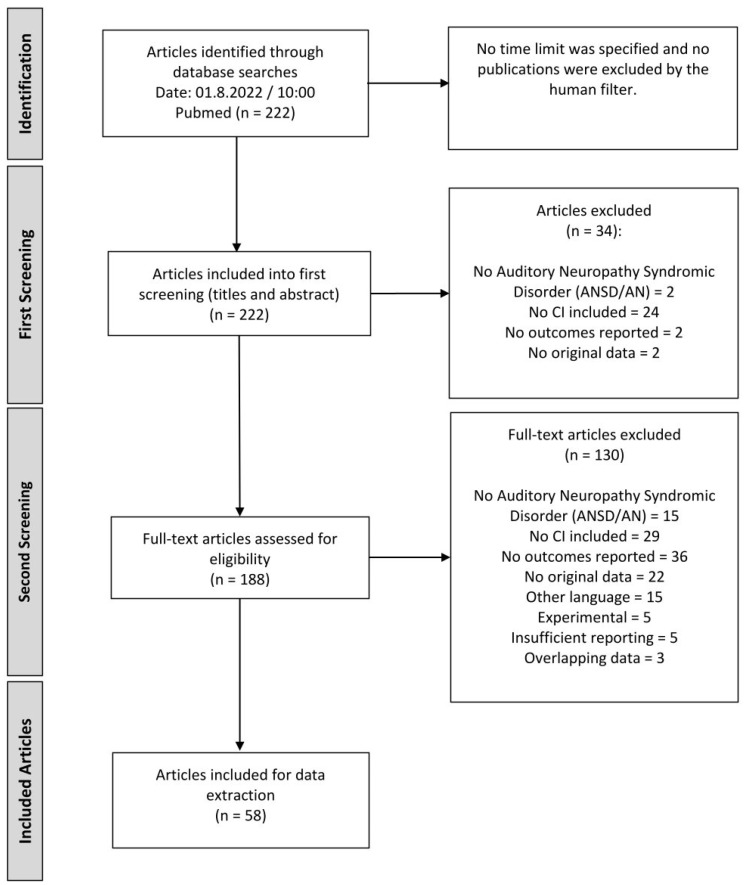
Flow diagram of study selection according to the PRISMA guidelines [43].

**Figure 3 jpm-13-00848-f003:**
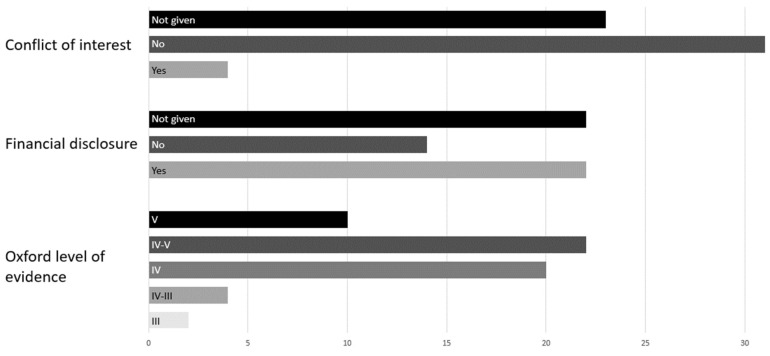
Report conflicts of interest, financial support and Oxford level of confidence of included publications.

**Figure 4 jpm-13-00848-f004:**
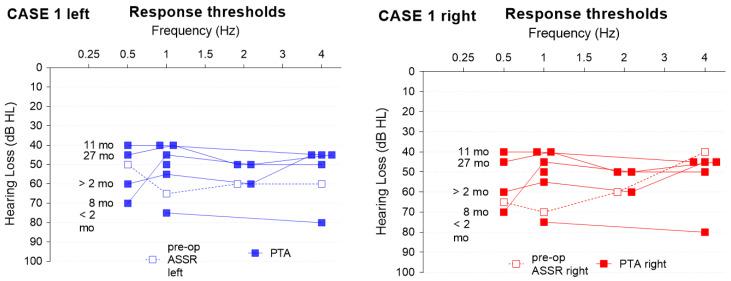
ASSR and pure tone results. Case 1: the follow-up time (months, mo) is denoted on the left. Post-op left/right hearing levels were identical.

**Figure 6 jpm-13-00848-f006:**
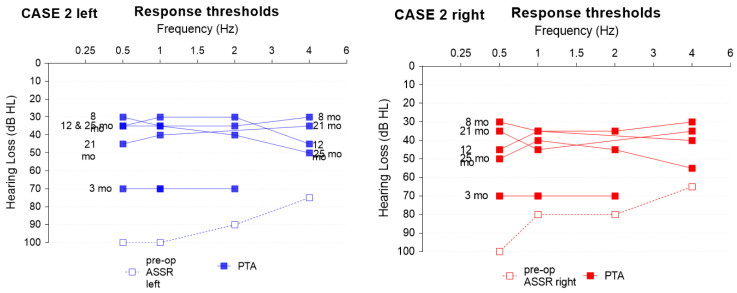
ASSR and pure tone results. Case 2; the respective follow-up time (months, mo) is denoted in the middle of the graph for the left and right side.

**Figure 7 jpm-13-00848-f007:**
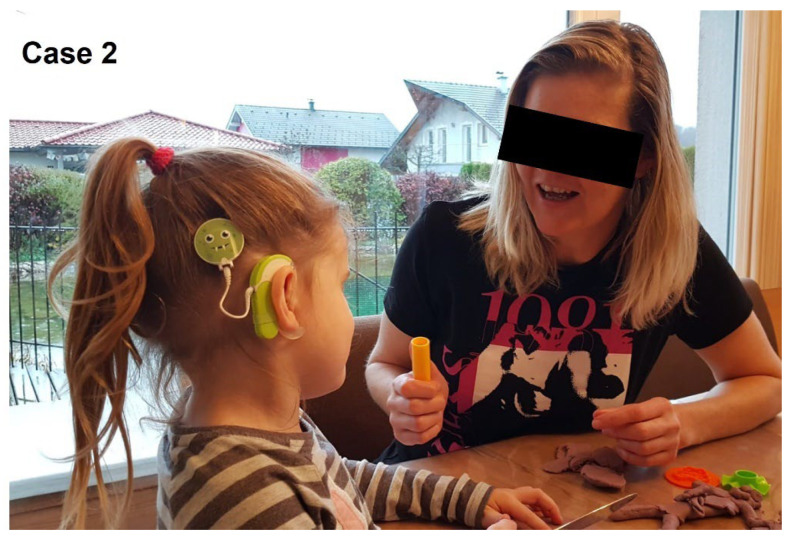
At home play training as part of the rehabilitation process with the mother.

**Table 1 jpm-13-00848-t001:** Identified PICOS to define inclusion and exclusion criteria for reviewed literature.

Inclusion Criteria	
**P**opulation	Children and adolescents of any gender or ethnicity with auditory neuropathy spectrum disorder.
**I**ntervention	Cochlear implant
**C**omparator	Not applicable
**O**utcomes	Data regarding audiological outcomes, language acquisition, general performance, quality of life, satisfaction, subjective outcomes.
**S**tudy design	Randomized or nonrandomized comparative studies, case reports, case series, case–control studies, controlled/not controlled before and after studies and interrupted time series analyses. Letters, editorials and systematic reviews with no original data, animal, in-vitro and laboratory studies were excluded.
Exclusion Criteria	
	Other language than English or German; publication lacking sufficient information for evaluation; overlap in data

**Table 2 jpm-13-00848-t002:** Search terms and achieved hits in the database PubMed.

Search Steps	Search Terms	Hits
1	Auditory Neuropathy AND Cochlear implant OR (Auditory Neuropathy AND Cochlear implant *)	222
2	Limit to Humans	222

* replaces one or more characters at the end of the respective search term to accept multiple forms including plurals

## Data Availability

Data available on request.

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
