# Peer review of "Systematic Literature Review and Early Benefit of Cochlear Implantation in Two Pediatric Auditory Neuropathy Cases"

_jpm, 2023, doi:10.3390/jpm13050848_

Round 1

Reviewer 1 Report

Children with auditory neuropathy spectrum disorder (ANSD) comprise an extremely heterogeneous group, as this disorder can result from sensopathy, synaptopathy, or neuropathy. As a result, ANSD manifests in various ways and requires a diverse range of habilitation options, including hearing aids and cochlear implants.

Therefore, the systematic literature review conducted by the authors of this manuscript provides valuable insights into the general trends of cochlear implant usage for children with ANSD and determines its effectiveness.

The relevance of this research is unquestionable, and its publication holds practical significance. The literature search and analysis are executed accurately, and the results are presented clearly. The article includes two excellent examples. With minor revisions as recommended below, the manuscript can be recommended for publication.

1.     Both the abbreviation ANSD and AN appear in the text. Use only one abbreviation throughout the text (ANSD is preferable).

2.     Introduce the abbreviation HL (hearing loss).

3.     Ensure references to figures are given in ascending order, e.g., consider moving the first mention references to Figures 3-5 to the Results section. Include a reference to Figure 2 (currently missing, an error message is given instead).

4.     It makes sense to give additional information about the children's medical history, the results of additional examinations (CT, MRI, genetic test etc.) given in the two examples. Possible etiology of ANSD are not indicated.

5.     It is not clear why Fig.5 left shows ASSR data for the right ear and vice versa, Fig.5 right contains ASSR data for the left ear.

6.     Consider the appropriateness of including Figure 6 (at the authors discretion).

Author Response

The authors would like to thank the reviewers for their thorough review of our manuscript entitled “Systematic Literature Review and Early Benefit of Cochlear implantation in two pediatric auditory neuropathy cases”, and their constructive suggestions to improve our research further. We have implemented all suggested changes in the document and re-submitted one version with track-changes active and one clean version. Please find respective comments and information of line and page number below.

Reviewer 1

  1. Both the abbreviation ANSD and AN appear in the text. Use only one abbreviation throughout the text (ANSD is preferable).

Changes were made throughout the manuscript (ANSD)

  1. Introduce the abbreviation HL (hearing loss).

Hearing loss was introduced in the first sentence in the Introduction (please see lines 25ff)

  1. Ensure references to figures are given in ascending order, e.g., consider moving the first mention references to Figures 3-5 to the Results section. Include a reference to Figure 2                (currently missing, an error message is given instead).

References to figures are in ascending order at first mention as per Journal guidelines (close to where they are discussed/mentioned). The results of the two cases are on purpose with the respective case as the outcome section is purely that of the Systematic Review.

  1. It makes sense to give additional information about the children's medical history, the results of additional examinations (CT, MRI, genetic test etc.) given in the two examples.

Details to the medical history were extended please see pg 12 lines 56 ff

  1. It is not clear why Fig.5 left shows ASSR data for the right ear and vice versa, Fig.5 right contains ASSR data for the left ear.

Case 1: outcomes were exactly the same for left and right – but figure was now adjusted the same as for case 2 – please see new Figure 4 rev and improved Figure 5 (pure left and pure right pre-op outcomes)

  1. Consider the appropriateness of including Figure 6 (at the authors discretion).

The authors are not sure what the reviewer is implying? Especially the mother of case 2 is very happy about the progress of her daughter and has voluntarily provided several pictures of their training sessions and play groups.

Reviewer 2 Report

Dear Editor,

I reviewed the article entitled “Systematic Literature Review and Early Benefit of Cochlear implantation in two pediatric auditory neuropathy cases” by Keintzel et al discussing the possible benefit of cochlear implantation in cases of auditory neuropathy.

The article in composed of two parts, one about two cases from the clinical experience of the authors and one about the literature review.

The article is well written (with small mistakes, that are addressed within the pdf of the text), the cases are presented well, and the study protocol is adequate to support the following statements.

The most interesting part is the literature review, which can support the proposal of cochlear implantation as a possible option for patients with auditory neuropathy.

So, in my opinion this paper can be published in the Journal of Personalized Medicine after some small revisions.

No comment

Author Response

The authors would like to thank the reviewers for their thorough review of our manuscript entitled “Systematic Literature Review and Early Benefit of Cochlear implantation in two pediatric auditory neuropathy cases”, and their constructive suggestions to improve our research further. We have implemented all suggested changes in the document and re-submitted one version with track-changes active and one clean version. Please find respective comments and information of line and page number below.

Reviewer 2

All changes suggested in pdf were made – please see accordingly page 2 Lines 78 and 80 ff, page 12 lines 41ff, line 252